# Emergence and spread of two SARS-CoV-2 variants of interest in Nigeria

Identifying the dissemination patterns and impacts of a virus of economic or health importance during a pandemic is crucial, as it informs the public on policies for containment in order to reduce the spread of the virus. In this study, we integrated genomic and travel data to investigate the emergence and spread of the SARS-CoV-2 B.1.1.318 and B.1.525 (Eta) variants of interest in Nigeria and the wider Africa region. By integrating travel data and phylogeographic reconstructions, we find that these two variants that arose during the second wave in Nigeria emerged from within Africa, with the B.1.525 from Nigeria, and then spread to other parts of the world. Data from this study show how regional connectivity of Nigeria drove the spread of these variants of interest to surrounding countries and those connected by air-traffic. Our findings demonstrate the power of genomic analysis when combined with mobility and epidemiological data to identify the drivers of transmission, as bidirectional transmission within and between African nations are grossly underestimated as seen in our import risk index estimates.

With a population of over 200 million, Nigeria is the most populous country in Africa[1]. Since the first report of SARS-CoV-2 in Nigeria on the 27th of February 2020, the cumulative number of confirmed COVID-19 cases in Nigeria has risen to more than 265 000 as of mid-September 2022[2]. However, this burden is extremely low relative to SARS-CoV-2 infections in the rest of the world. Adjusting for population size, there have been 129 cases per 100,000 people in Nigeria as of mid-September, compared to more than 34,700 and 28,700 in the UK and USA, and 2,340 and 711 in Indonesia and Pakistan, respectively. Concurrently, Nigeria has also only had a little over 3,000 reported deaths (2 per 100,000), compared to the UK and USA, which had 303 and 315 cumulative deaths per 100,000 people[2]. This relatively low but heterogeneous incidence in Nigeria and across the wider African region has been the cause of persistent speculation, including on the putative central role of case underascertainment[3–5]. The underlying cause remains understudied and is likely multifactorial, with many other factors speculated to contribute, including skewed age structures of populations, more restricted human mobility in certain regions, host genetics, environmental factors, and potential pre-existing population immunity to related viruses[6,7]. Before the emergence of globally sweeping variants such as Delta and Omicron, it was also unclear if the specific genetic diversity circulating in the African region may have contributed to the heterogeneous incidence and mortality observed[8–10]. It is therefore important to characterise the genomic epidemiology of SARS-CoV-2 in Nigeria over the course of the pandemic, to improve our understanding of what variants emerged to dominate the different epidemic waves. It is also important to improve our understanding of the drivers of transmission in the region, as this remains understudied in Africa and findings from other regions may not be generalizable[9,11]. Nigeria is highly connected to its neighbouring countries and the wider African region, potentially acting as a dominant source of transmission via the high volume of movement across land as well as air borders[12].

Several SARS-CoV-2 variants, which have a constellation of mutations that are biologically significant to the virus, emerged during the pandemic[9,13,14]. The B.1.525 (Eta) and B.1.1.318 variants of interest (VOIs) dominantly co-circulated with the Alpha variant during the second wave in Nigeria from December 2020 to March 2021[2]. B.1.525 and B.1.1.318 were suggested to have emerged in Nigeria based on epidemiological reports from travellers in countries such as the UK, India, Mauritius, Canada, and Brazil[15–19]. Notably, B.1.525 and B.1.1.318 both showed a significant increase in the infectivity of human and monkey cell lines experimentally, raising concerns of intrinsic

✉ e-mail: happic@run.edu.ng

increased transmissability[20]. Both of these lineages also share the well-characterised E484K substitution in the Spike protein receptor-binding domain (RBD), which effectively reduces antibody neutralisation[21].

In this study, we examine the genomic epidemiology of SARS-CoV-2 in Nigeria from March 2020 to September 2021, across the first three epidemic waves. In particular, we investigate the timing and origin of the emergence of the B.1.525 and B.1.1.318 VOIs. In phylogeographic reconstructions, we characterized the source-sink profiles of these best-sampled VOIs to better understand the bidirectional transmission dynamics of SARS-CoV-2 in Nigeria and the wider African region. We compare our findings from our genomic data to integrated travel and epidemiological data to explore the role of Nigeria's regional and intercontinental connectivity in the estimated import and export dynamics.

## Results

### Lineage dynamics of SARS-CoV-2 in Nigeria over the first three epidemic waves

We generated a total of 1577 genomes from samples obtained between March 2020 and September 2021, across the first three waves of the pandemic (Fig. 1A, B). We collected samples from 25 of the 36 states and the Federal Capital Territory of Nigeria (Fig. 1C). The northern states were highly undersampled relative to the southern states, with sampling highly uneven across time, especially during the third wave (Fig. 1A, B).

To investigate the lineage dynamics across the different outbreak waves, we used the pangolin nomenclature tool to assign all sequences to corresponding lineages. We detected more than 35 lineages across the first three waves (Fig. 1D). We found that a number of ancestral lineages circulated during the first wave, including A, B, B.1, and B.1.1

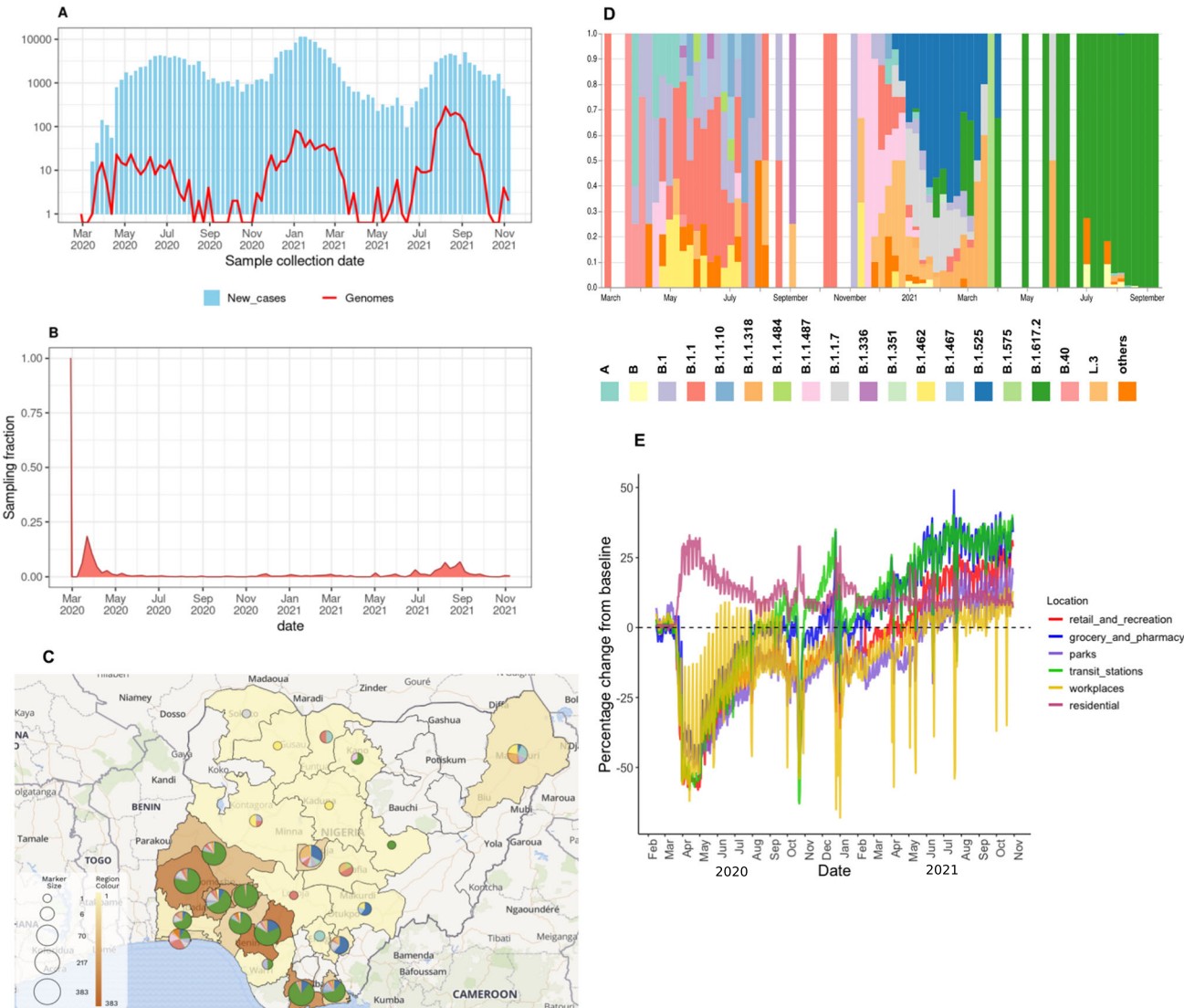

**Fig. 1 | Genomic epidemiology of SARS-CoV-2 in Nigeria. A** Epicurve of SARS-CoV-2 cases in Nigeria in Nigeria and genomes produced assembled during the first three waves with the y-axis transformed to a log-scale. **B** Time-varying sampling fraction of genomes produced in this study per new cases. **C** Geographic distribution of sequences generated in the current study. The map of Nigeria shows the number of genomes from states across the country as region colour and marker size display lineages per state. Maps © Mapbox (www.mapbox.com/about/maps) and © OpenStreetMap (www.openstreetmap.org/about). **D** Lineage frequency profile for the study period. The frequency of PANGO lineages across the country over a period of 80 weeks. Lineages that are not VOCs or VOIs, or that do not appear more than three times over the course of the pandemic, are grouped as "others". **E** Human mobility data from Google for retail shops, pharmacies, parks, public transport, workplaces, and residential activity places from January 2020 to November 2021, as compared to the pre-pandemic baseline (January to February 2021).

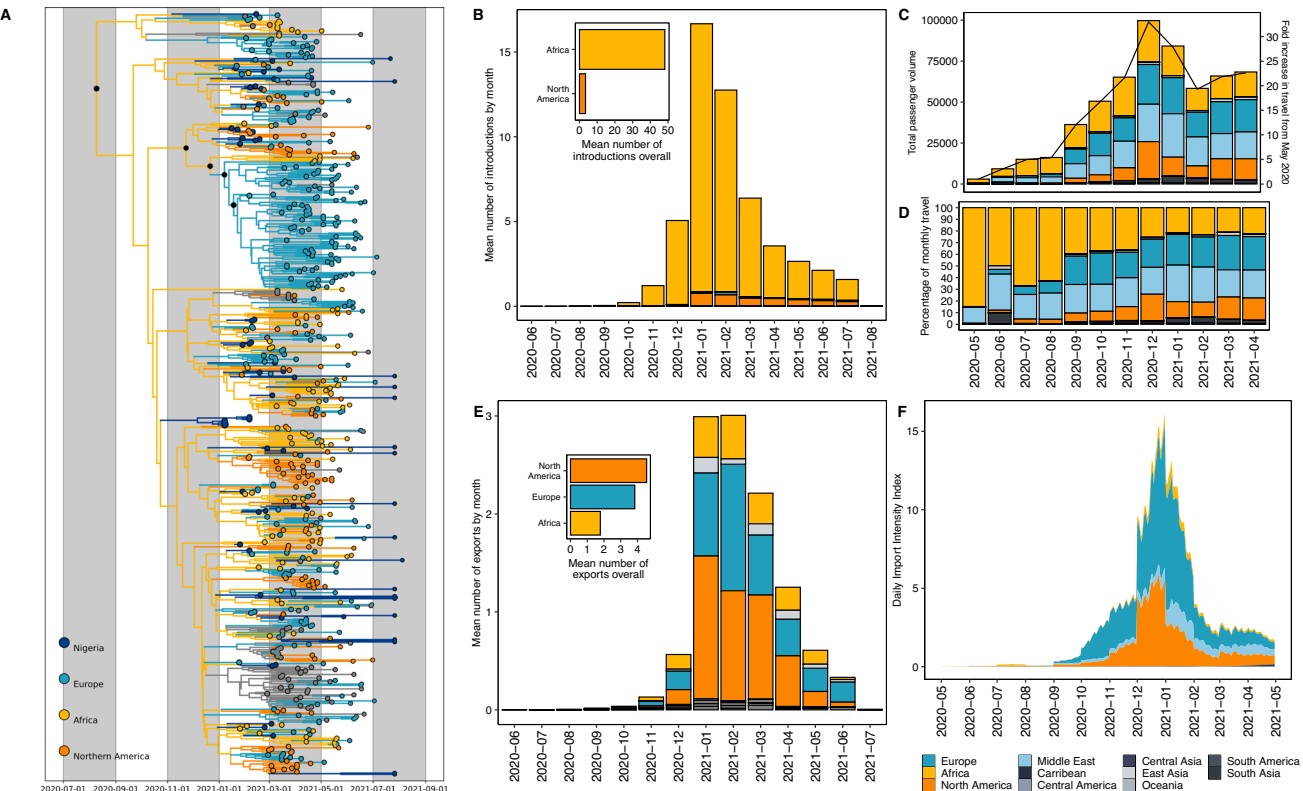

**Fig. 2 | Bayesian phylogeographic reconstruction of the B.1.1.318 variant.**
**A** Time-resolved B.1.1.318 phylogeny. Branches are coloured by region-level geographic state reconstruction. Locations with negligible contributions have been grouped and annotated in grey. Internal nodes annotated with black points represent posterior support > 0.75. **B** Mean number of introductions (Markov jumps) of B1.1.318 into Nigeria from all source regions, binned by month. The mean number of introductions overall (inset). **C** Volume of monthly air passengers inbound to Nigeria across 2020–2021 by source region (bar, left axis) and fold increase in travel volume compared to May 2020 baseline. **D** Estimated Introduction Intensity Index for Nigeria from 2020 to 2021. **E** Percentage of inbound air travel to Nigeria by source region across 2020–2021. See Supplementary Fig. 2 for country-level data. **F** Estimated Importation Intensity Index (III) for Nigeria from 2020 to 2021.

(Fig. 1D). The onset of community transmission in Nigeria was initially delayed, with the first wave initiated two months after the first case detection in March 2020. A nationwide lockdown and ban on local and international air travel were enforced, with the first wave ending in August 2020 after four months (Fig. 1B). During the lockdown, we found that there was up to a 50% decline in activity at workplaces, retail, supermarkets, and public transport stations, and up to a 25% increase in residential activity (Fig. 1E).

After this strictly adhered-to lockdown, Nigeria opened its airspace to international traffic in October 2020. This was shortly followed by an increase in new cases in November 2020, marking the start of the second wave (Fig. 1B), which was characterised by B.1.1.7, B.1.525, B.1.1.318 (VOC and VOI), and other variants such as L.3 and B.1.1.487 (Fig. 1D).

The following analyses focused on the lineages (B.1.525 and B.1.1.318) that contributed to the surge during the second wave and have also been shown to possess genotypic traits for increased transmissibility and also increased virulence in human and animal cell lines, such as the E484K spike protein RBD mutation[21]. We excluded B.1.1.7 from our analysis as it has already been well studied[22,23], and it was not the dominant variant in Nigeria during the second wave, unlike in other places where it was reported, such as the UK and the USA[24–26]. Moreso, the second wave began after travel restrictions were lifted and human mobility was back to normal, thus the need to investigate the emergence of B.1.1.318 and B.1.525 in Nigeria. The third wave was initiated in June 2021, with the Delta variant (B.1.617.2) and its sub-lineages sweeping to dominance in sampling (Fig. 1B, D). In the other analysis described in this work, we

also sought to identify the timings of emergence and number of introductions of these VOI.

## Regional connectivity drove the introduction of B.1.1.318 into Nigeria

Using Bayesian phylogeographic reconstructions, we investigated the timing and origin of the B.1.1.318 emergence to better understand Nigeria's connectivity in the global SARS-CoV-2 transmission networks (see Methods). B.1.1.318 was first detected in Nigeria in Lagos State in December 2020. The lineage was detected in multiple countries within and outside Africa, notably resulting in a large outbreak in Mauritius[19].

In our phylogeographic reconstruction, we found that B.1.1.318 emerged in Africa (root state posterior support = 0.95-6) in early August 2020 [mean tMRCA = 5 August, 95% HPD 25 June to 20 September] across two replicates (Fig. 2A). We estimated that B.1.1.318 was introduced to Nigeria on at least 53 independent occasions [mean introductions, 95% HPD 50-59] beginning in November 2020, after travel restrictions were lifted in October 2020 (Fig. 2B). We found that the majority of introductions originated from other African nations (Fig. 2B), indicating Nigeria's strong connectivity to the region. However, the number and origin of introductions and exports estimated from genomic data are sample dependent, with bidirectional transmission to and from undersampled regions obscured by uneven sampling globally. We quantified air travel patterns to and from Nigeria over time to better understand which countries were most likely to act as transmission sources or sinks based on their connectivity to Nigeria.

Before travel restrictions were lifted in October 2020, we found that the low levels of incoming air travel volume predominantly

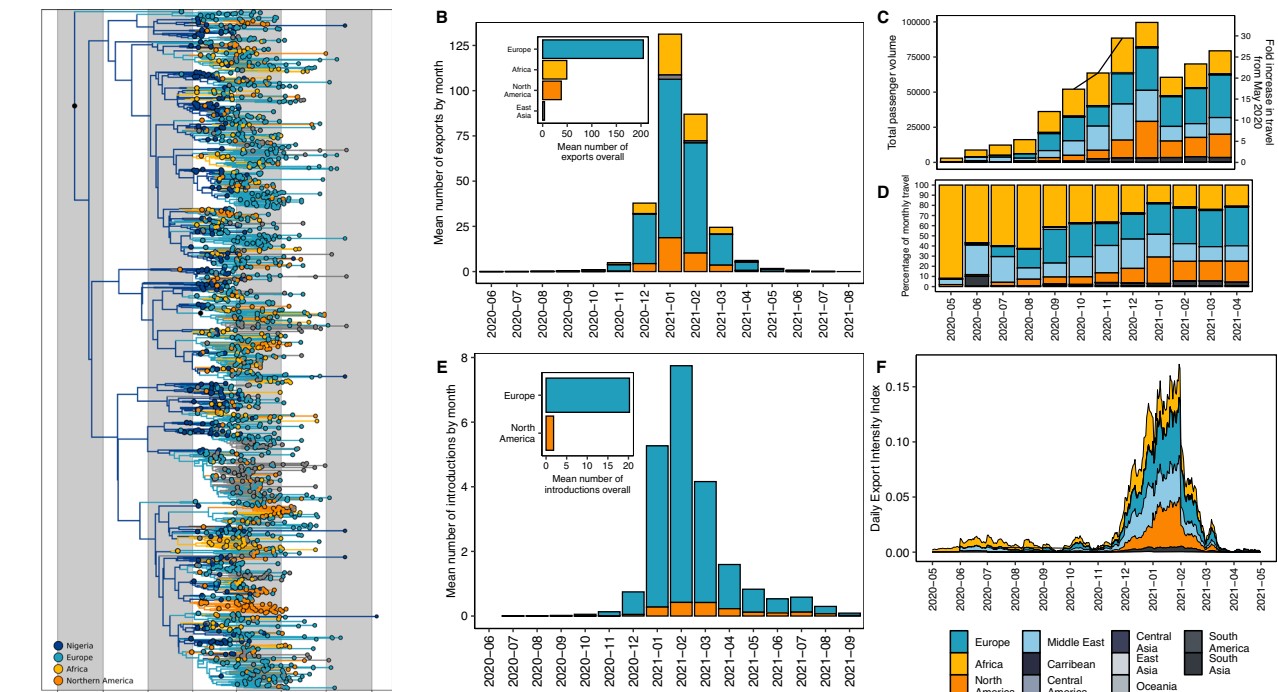

**Fig. 3 | Bayesian phylogeographic reconstruction of the B.1.525 variant. A** Time-resolved B.1.525 phylogeny. Branches are coloured by region-level geographic state reconstruction. Locations with negligible contributions have been grouped and annotated in grey. Internal nodes annotated with black points represent posterior support > 0.75. **B** Mean number of exports (Markov jumps) of B.1.525 from Nigeria to destination regions, binned by month. Mean number of exports overall (inset). **C** Volume of monthly air passengers outbound from Nigeria across 2020–2021 by destination region (bar, left axis) and fold increase in travel volume compared to baseline (May 2020). **D** Percentage of outbound air travel from Nigeria by destination region across 2020–2021. See Supplementary Fig. 1 for country-level data. **E** Mean number of introductions (Markov jumps) of B1.525 into Nigeria from all origin regions, binned by month. The mean number of introductions overall (inset). **F** Estimated Exportation Intensity Index (EII) for Nigeria from 2020 to 2021.

originated from other African nations, followed by the Middle East (driven by the UAE) and Europe (driven by the UK) (Fig. 2C,D, Supplementary Fig. 2). This suggests that the introduction risk was predominantly driven by regional connectivity during the period when travel restrictions were in place. We sought to investigate this further by quantifying the introduction risk for all countries connected to Nigeria by air travel. We estimated the introduction intensity index (III), which accounts for the number of cases in the source countries as well as the travel volume to Nigeria (see Methods). Overall, we found that the introduction risk from air travel was very low during the period of travel restrictions (May 2020–October 2020), as incoming travellers originated largely from other African nations with low incidence (Fig. 2F).

We found that the number of incoming air passengers peaked in December 2020–January 2021 (Fig. 2C, D). We also observed that the number of inbound passengers from Europe, the Middle East, and North America increased relative to other African nations after restrictions were lifted, resulting in comparable volumes from Africa, Europe, and the Middle East (Fig. 2C, D). The 30-fold increase in incoming passengers by December 2020 was reflected in the corresponding peak in the III, attributable to increased introduction risk from Europe and North America from November 2020 into early 2021 (Fig. 2F). The surge was predominantly driven by the high-incidence second waves of the UK and USA, respectively (Supplementary Fig. 1).

We found that the III peak but not the estimated sources of risk were consistent with the number and origin of introductions of B.1.1.318 in our phylogeographic reconstructions (Fig. 2B). We estimated the highest introduction risk for Europe after travel restrictions were lifted, whereas the majority of introductions estimated from genomic data originated in other African countries. The III from other African nations was relatively low, despite comparable incoming air

travel volume. This is likely both a factor of the relatively low incidence in most African countries during this period (Fig. 2F, Supplementary Fig. 3) as well as the fact that our travel data does not account for land-based travel, which will expectedly severely underestimate the introduction risk from surrounding countries. The III from the African region was largely driven by South Africa, which was experiencing a peaked incidence from its second wave (Supplementary Fig. 3). In our III analyses, we found negligible introduction risk from outside of Europe and North America.

## The Eta variant of interest (B.1.525) likely emerged in Nigeria

The B.1.525 (Eta) lineage drove the second wave from January to March 2021. B.1.525 was initially reported to have originated in Nigeria or the UK[16–18]. We investigate the timing, origin, and transmission dynamics of B.1.525 with Bayesian phylogeographic reconstructions (see Methods). We estimated that the B.1.525 lineage emerged in Nigeria (root state posterior support = 0.998) in late July 2020 [mean tMRCA 23 July, 95% HPD 7 June to 4 September], during the final weeks of Nigeria's first wave (Fig. 3A). This suggests that this lineage circulated cryptically for several months before its first detection in Nigeria on 12 December 2020, likely owing to low-incidence-associated sparse sampling (Fig. 1A).

We sought to better understand Nigeria's connectivity in global SARS-CoV-2 transmission networks by identifying the most likely destinations of B.1.525's exportation, as the lineage (1) emerged in Nigeria and (2) is Nigeria's most sampled lineage, excluding the Delta variant sublineages. To investigate the spread of B.1.525 from Nigeria, we reconstructed the timing and pattern of geographic transitions out of and into Nigeria across the full posterior of our Bayesian phylogeographic reconstructions. We estimated that B.1.525 was exported from Nigeria a lower bound of 295 times [mean Markov jumps, 95%

HPD 259-335] (Fig. 3B). The majority of sampled exports were destined for Europe, followed by the rest of Africa and Northern America from December 2020 to March 2021 (encompassing Nigeria's second wave) (Fig. 3B). During this period, we also found that B.1.525 was re-introduced from Europe a lower bound of 20 [95% HPD 6-37] times (Fig. 3E). These estimated source-sink dynamics support Nigeria's strong connectivity, particularly to Europe, but will underestimate exports to undersampled regions such as neighbouring African countries. To mitigate the effect of global sampling biases on genomic estimates of transmission dynamics, we again analysed changes in air travel to and from Nigeria over time to understand the sources and seeds of Nigeria's bidirectional transmission dynamics.

African nations dominated the reduced level of outbound travel during the period of travel restrictions (May to October 2020) (Fig. 3C, D), suggesting that regional connectivity drove Nigeria's export risk during this period. We integrated the air travel data with Nigeria's epidemic incidence data to quantify an exportation intensity index (EII)[27]. The EII quantifies the temporal trend in the daily estimated number of viral exports from Nigeria to sink countries (see Methods). Given the data available, Nigeria's estimated export intensity was low overall compared to countries with high air traffic data to Nigeria, as the country's SARS-CoV-2 incidence remained comparatively low on a global-scale for the entire period under investigation (Fig. 3F). Nigeria's first wave had relatively low incidence, peaking at about 4000 cases in a week in late June (week 18) (Supplementary Fig. 4B). Combined with reduced travel volumes, the period from May 2020 to September 2020 was characterised by negligible export intensity (Fig. 3F).

Outbound air travel recovered gradually over 2020, peaking in December 2020–January 2021 (Fig. 3C, D). We found that this peak corresponded with the peak of the larger second wave in Nigeria and therefore also the EII, as well as the distribution of exports of B.1.1.525 from Nigeria (Fig. 3B, F). The proportion of outbound air travel destined for countries in Europe, the Middle East, and North America increased relative to travel destined for African countries after travel restrictions were lifted (Fig. 3C, D). We estimated the highest EII for Europe (driven predominantly by travel to the UK) and Northern America (driven by travel to the USA) from December 2020 to February 2021 (Supplementary Fig. 4). The export intensity to Europe is consistent with the high number of estimated exports in the genomic data, though North America was disproportionately underestimated in the genomic data (Fig. 3B). Overall, air travel volume destined for Europe was only moderately higher (10%) than Africa, North America, and the Middle East from December (after first detection and at the start of the second wave) to April 2021 (end of the second wave) (Fig. 3C, D). However, Europe had 4-fold more sampled introductions of B.1.525 than African nations or the USA. Notably, the comparatively high EII for the Middle East region (representing 20% of outbound volume overall) was not represented in our phylogeographic reconstructions (Fig. 3B). This highlights how unevenly distributed surveillance capacity can lead to an underestimation of transmission events (Supplementary Fig. 3A).

## Discussion

In this study, we combined genomic, travel and epidemiological data to characterise the emergence and bidirectional transmission of two focal variants of interest in Nigeria. The focal VOIs, B.1.525 and B.1.1.318, were suggested to have emerged in Nigeria before spreading globally, resulting in large-scale outbreaks in Brazil and Mauritius[15,19]. In our phylogeographic reconstructions, we found that the B.1.1.318 lineage most likely emerged in the African region in early August 2020, with its dominance in Nigeria driven by multiple introductions during the second wave from November 2020 onwards. We also found that the Eta variant (B.1.525) emerged in Nigeria in late July, with high levels of export to Europe reflected in the genomic and estimated export index.

The true extent of bidirectional transmission with other African nations will be severely underestimated in phylogeographic reconstructions, as Africa remains disproportionately undersampled despite heroic efforts[28]. Our findings should be interpreted in the light of our own sampling fraction, which was 0.026% of reported cases. We attempted to mitigate these global surveillance biases by supplementing genomic estimates with sampling-independent metrics like the introduction and export intensity indices. The III and EII indices should be interpreted to quantify the temporal trend in the daily estimated number of introductions or exports rather than absolute values. The indices are based on the back-extrapolated time series of deaths and are therefore restricted by the associated reporting delays and biases. Notably, they will be underestimated if there is large-scale underascertainment in source countries, as previously shown for African nations, including Nigeria, based on excess mortality data[29–31]. Most notably, our import risk index underestimates the importance of regional connectivity in driving introduction estimates, as they likely result from shorter distance connectivity to surrounding countries, which we could not collect data on. One of the biases recorded in our study was that we were not able to distinguish samples based on travel and community testing due to insufficient metadata. Hence, a sensitivity analysis to exclude travel-based testing was not possible. Another limitation is the inability to sequence samples from all the states in Nigeria at the same proportion across time, as the northern states were undersampled during the third wave. Hence, the within-country dynamics could not be determined accurately due to complications that arose relating to sample transfer and data sharing from other partnering labs during the pandemic. This also calls for better data-sharing agreements and guidelines for genomic surveillance of pathogens in Nigeria where trust is a big issue.

At an unprecedented speed (72 h from receiving the sample), ACEGID generated the first whole-genome sequence of the virus in Africa in March 2020. This catalysed and built confidence in SARS-CoV-2 genomic sequencing on the African continent. This immense collaborative effort, involving seventeen partner laboratories in Nigeria, helped to characterise the pandemic in near real-time during its first three waves in the country. As we have seen in recent years, novel variants of interest or concern can emerge from anywhere around the world, notably including undersampled regions such as Africa[13,14]. In a highly connected world, there is a proactive need to adopt early warning systems for pandemic preemption and response in Africa, such as SENTINEL[32]. This would enable equality in global genomic surveillance, especially in countries with fragile public health systems, in order to effectively detect and curb emerging outbreaks before they spread.

## Methods

### Study population/sampling

The study was approved by the National Health Research Ethics Committee of Nigeria (NHREC), with protocol number NHREC/01/01/20017-08/08/2020 and approval number NHREC/01/01/2007-30/11/2021B. Informed written consent was obtained directly from the patient as part of the routine surveillance program in Nigeria, therefore there was no participant compensation. Samples were collected from people who reported to community testing centres for COVID-19 tests (travellers included) and hospitalised individuals from February 2020 to October 2021 across the country. Sampling fraction was deduced by the number of SARS-CoV-2 genome(s) assembled against the number of confirmed cases in a given day.

### Sample processing

RNA was extracted from nasopharyngeal swabs in viral transport media and saliva/sputum in PBS using the QiAmp viral RNA mini kit (Qiagen, Hilden, Germany) according to the manufacturer's instructions and MagMax pathogen RNA/DNA kit (Applied Biosystems,

Massachusetts, USA) using a Kingfisher Flex purification system (ThermoFisher Scientific, Massachusetts, USA) according to the manufacturer's instructions. RT-qPCR screening of suspected samples was carried out targeting N, ORF1ab and RdRP genes of the virus using commercially available kits: genesig® (Primerdesign Ltd, UK), DaAnGene (Daan Gene Co., Ltd, China), Liferiver (Shanghai ZJ Bio-Tech Co., Ltd, China), Genefinder (OSANG Healthcare Co., Ltd, Korea), and Sansure (Sansure Biotech Inc., China).

## SARS-CoV-2 whole-genome sequencing

Random samples with moderate to high viral load detected with RT-qPCR (Ct value <30) were selected for sequencing, which were samples collected from routine sentinel surveillance and hospitalised individuals spatially and temporally. These are people who: (i) had COVID-19 symptoms and reported to testing centres, (ii) were hospitalised or under quarantine due to COVID-19 related complications, (iii) are inbound or out-bound travellers who had to take mandatory COVID-19 tests, or (iv) were tested based on reported outbreaks within the community. Samples were collected from both private and public COVID-19 testing labs across the country in twenty-six states (Adamawa, Akwa Ibom, Benue, Borno, Delta, Ebonyi, Edo, Ekiti, Enugu, FCT Abuja, Kaduna, Kano, Katsina, Kogi, Kwara, Lagos, Nasawara, Niger, Ogun, Ondo, Osun, Oyo, Plateau, Rivers, Sokoto, and Zamfara states). The sampling gap was the inability to sequence samples from every state during all different waves of the pandemic in Nigeria. The Illumina COVIDseq protocol was used for sequencing preparation (https://www.illumina.com/products/by-type/clinical-research-products/covidseq-assay.html). RNA was converted to cDNA, followed by tiling amplification (400 bp) with Artic V3 primer pools, which covers the whole genome of the virus. Nextera DNA flex libraries were made from the amplicons. Libraries were pooled and sequenced on the Illumina MiSeq, NextSeq 2000, and NovaSeq 6000 platforms at the African Center of Excellence for Genomics of Infectious Diseases (ACEGID), Redeemer's University, Ede, Nigeria.

## Genome assembly

We used the viral-ngs pipeline v2.1.19 (https://github.com/broadinstitute/viral-ngs) for demultiplexing, quality control, and genome assembly, consistent with prior work[33–35]. The demux_plus applet of viral-ngs was used for demultiplexing basecalls to BAM files. Genome assembly was carried out using the assemble_refbased workflow on the viral-ngs pipeline which maps each unmapped BAM file to the SARS-CoV-2 reference genome (NC_045512.2) to generate coverage plots and FASTA files of successfully assembled genomes.

## Lineage/clade assignment

We used Phylogenetic Assignment of Named Global Outbreak LINeages (PANGOLIN)[36] v3.1.12 to identify the lineages of SARS-CoV-2 circulating within Nigeria and to identify the lineages circulating in specific states of Nigeria. Nextclade v1.3.0 (https://clades.nextstrain.org/) was used to assign the sequences to globally circulating viral clades and to investigate mutations in the genome that could affect primer amplification during PCR. We visualised the lineage dynamics across Nigeria using Microreact[37] and a GeoJSON file of the map of Nigeria.

## Maximum likelihood phylogenetic analysis

1577 genomes with sequence length ≥20,930 (covering 70% or more of the reference−NC_045512.2) were assembled and aligned using MAFFT[38] v7.490. We used IQTREE[39] v2.1.2 and the GTR (generalised time-reversible) model with a bootstrap value of 1000 to construct the phylogenetic tree. Treetime[40] v0.92 was used for quantifying the molecular clock and identifying ancestral phylogeny in the augur pipeline of Nextstrain[41] v3.0.3.

## Bayesian phylogenetic analysis

All B.1.1.318 (n = 3858 with major locations: Europe, USA, and Africa accounting for 94%) and B.1.525 (n = 8278 with major locations: Europe, USA, and Africa accounting for 86%) sequences were downloaded from GISAID[42] on 18 August 2021. Sequences with >5% ambiguous nucleotides, a length <95%, or incomplete dates were discarded. Sequences were aligned to the reference (NC_045512.2) using minimap2, with the 5' and 3' UTRs and known problematic sites (GitHub−W-L/ProblematicSites_SARSCoV2 3) masked[43]. Both lineage-specific datasets were downsampled for representativeness and computational tractability with a phylogenetic-informed downsampling scheme. A maximum-likelihood phylogeny was reconstructed for each lineage with automatic model selection in IQTREE[44]. The phylogeny was downsampled by root-to-tip tree-traversal, with all internal nodes subject to two rules: (1) if 95% of the leaves subtended by the internal node represented a single country, the earliest representative was retained alongside a random sequence; (2) if leaves from the same location were separated by a zero-branch length, the earliest representative was retained alongside a random sequence. All countries with more than the median number of sequences across countries were then downsampled randomly to the median. This yielded a total number of 1118 and 1746 genomes for B.1.1.318 and B.1.525 respectively. All Nigerian sequences were retained (n = 73 for B.1.1.318; n = 256 for B.1.525).

We reconstructed Bayesian time-scaled phylogenies for each lineage using BEAST v1.10.5[45]. For both lineages, the time-scaled phylogenies were reconstructed under an HKY substitution model with a gamma-distributed rate for variation among sites[46], a relaxed molecular clock with a log-normal prior[47], and an exponential growth coalescent tree prior[48]. For each dataset, we combined two independent Markov Chain Monte Carlo (MCMC) chains of 200 million states, run with the BEAGLE package[49] to improve run time. Parameters and trees were sampled every 20,000 steps, with the first 20% of steps discarded as burn-in. Convergence and mixing of the MCMC chains were assessed in Tracer v1.7, to ensure the effective sample size (ESS) of all estimated parameters was > 200[50].

We performed an asymmetric discrete trait analysis using BEAST version 1.10.5 to reconstruct the location-transition history across an empirical distribution of 4000 time-calibrated trees (sampled from each of the posterior tree distributions estimated above). We aggregated the country of sampling on a regional level for computational tractability. We used Bayesian stochastic search variable selection (BSSVS) to infer non-zero migration rates and identify the statistically supported transition routes into and out of Nigeria by a Bayes factor test[51]. In addition to the discrete trait analysis, we used a Markov jump counting approach to estimate the timing and origin of geographic transitions into Nigeria to account for uncertainty in phylogeographic reconstruction associated with sparse sampling and low sequence variability[52]. We used the TreeMarkovJumpHistoryAnalyzer from the pre-release version of BEAST v1.10.5 to extract the Markov jumps from posterior tree distributions[53]. We used TreeAnnotator v1.10 to construct Maximum clade credibility (MCC) trees for all datasets. Trees were visualised using baltic (https://github.com/evogytis/baltic).

## Air traffic and human movement data

In order to associate international air travel and local human movement with variant movement across borders, we used air traffic data from the International Air Transportation Association (IATA) obtained from BlueDot (https://bluedot.global/) to quantify the volume of international travellers to and from Nigeria. The dataset included the number of travellers by origin and destination country, aggregated by month, across all international airports in Nigeria. Data was obtained for the period May 2020 to April 2021, encompassing the emergence and spread of B.1.525 and B.1.1.318. We also curated human movement

data within Nigeria from Google Mobility (https://www.google.com/covid19/mobility/) from January 2020 to November 2021, which reflects a baseline before lockdown (January–March 2020) and the study's period of interest. We used R to display movement patterns after grouping human movement into six divisions: retail & recreation, grocery & pharmacy, parks, transport stations, workplaces, and residential areas.

### Estimated introduction and exportation intensity index

From genomic data, estimates of the number and origin/destination of bidirectional transmissions with Nigeria are dependent on the sample's representativeness of the population and are therefore limited by global and local sampling biases. To limit these sampling biases, we supplemented our phylogeographic analyses by estimating the introduction (III) and export intensity index (EII) according to Du Plesis et al. [27]. The III estimates the daily risk of introductions into Nigeria from each country as a product of the number of asymptomatically infected individuals in each source country on that day (estimated from the time series of deaths) and their likelihood of travelling to Nigeria (based on the volume of inbound air travel from the source country). The EII is calculated similarly, based on the number of asymptomatically infected individuals in Nigeria and their likelihood of travel to each destination by air. Notably, we could not obtain data for land-based travel. Connectivity to regional and neighbouring countries and the associated introduction and export intensity are therefore likely severely underestimated. A time series of reported deaths for each country were collected from the outbreak.info R package [54]. As air travel data was aggregated by month, we conservatively assumed that travel was uniform across all days of the month. We quantified the III and EII for the period May 2020 to April 2021, encompassing the emergence and spread of B.1.525 and B.1.1.318.

### Reporting summary

Further information on research design is available in the Nature Portfolio Reporting Summary linked to this article.

## Data availability

The datasets and associated metadata used in this study are available in GISAID's EpiCoV database under the EPI_SET_ID accession numbers EPI_SET_221227vp (https://doi.org/10.55876/gis8.221227vp) and EPI-SET_221227 pc (https://doi.org/10.55876/gis8.221227pc). XML files used for the BEAST analysis are deposited in GitHub: https://github.com/acegid/SARS-CoV-2_Manucript_Supplemental_Data. The SARS-CoV-2 consensus genome assembly data generated in this study have been deposited in NCBI GenBank database under the accession numbers: OQ050230 to OQ052977 (https://www.ncbi.nlm.nih.gov/nuccore/?term=OQ050230:OQ052977[accn]). DNA sequences have been deposited in NCBI SRA under the BioProject PRJNA916503. Source data are provided as a Source Data file. Reference genome Wuhan-Hu-1 available in GenBank under accession MN908947.3. Air travel data can be requested for release from Bluedot (info@bluedot.global), with use pending approval by Bluedot. Source data are provided with this paper.

## Code availability

The scripts used for analysis reported in this study are publicly available at https://github.com/acegid/SARS-CoV-2_Manuscript_Supplemental_Data/tree/main/Figures [55].

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

## Acknowledgements

We gratefully acknowledge all data contributors, i.e., the Authors and their Originating laboratories responsible for obtaining the specimens, and their Submitting laboratories for generating the genetic sequence and metadata and sharing via the GISAID Initiative, on which this research is based. This work is made possible by the support provided to ACEGID by a cohort of generous donors through TED's Audacious Project, including the ELMA Foundation, MacKenzie Scott, the Skoll Foundation, and Open Philanthropy. This work was also partly supported by grants from the National Institute of Allergy and Infectious Diseases (https://www.niaid.nih.gov), NIH-H3Africa (https://h3africa.org) (U01HG007480 and U54HG007480), the World Bank (projects ACE-019 and ACE-IMPACT), the Rockefeller Foundation (Grant #2021 HTH), the Africa CDC through the African Society of Laboratory Medicine [ASLM] (Grant #INV018978), the Wellcome Trust (Project 216619/Z/19/Z), the WARN-ID/CREID grant # U01AI151812, and the Science for Africa Foundation. We want to especially thank the Redeemer's University Management, the State Government of Osun and NCDC for their support during the course of the COVID-19 pandemic.

## Author contributions

I.B. Olawoye, P.E.O., and E.P., conceived the study. C.T.H., P.C.S., K.G.A., obtained the funding. J.U.O., J.N.N., A.T.K., T.J.O., F.V.A., I.B. Olawoye, P.E.O., developed the methodology, sequencing, and wrote the original draft. P.E.E., O.A.F., A.N.H., O.T., B.L.S., C.I., I.M.A., M.M.B., F.O., S.F.S., N. Ndodo, I.N., R.A., coordinated the study. P.A., T.A.S., C.A.U., U.E.G., F.A., K. Akano, N.E.O., I.F., K. Adedotun-Sulaiman, F.B.B., B.B.A., C.P., R.A.A., G.C.C., M.I.A., O.O.O., S.G.O., O.A.O., M.F.S., A.E.S., G.O.E., O.G.J., J.O.A., O.O. Akinlo, O.O.F., T.O.I., D.C.N., A.E.O., I.B. Omwanghe, C.A.T., J.O., O. Ayo-Ale, O.I., E.B., G.O.N., A.E.P., O. Blessing, A.M., A.J., J.O.A., P.E., O.R., E.R., G.E.R., E.S., E.A., Y.E., A.O.C., A.I.D., E.O., M.Y.T., H.E.O., M.B., R.A.A., C.K.O., J.O.S., A.O., A.E.A., A.B., F.D., I.F.Y., A. Fajola, N. Ntia, J.J.E., A.E.M., B.W.M., O.E.F., M.A., I.M.K., B.S.O., Z.W.W., O.O. Adeyemi, O.A.A., A. Ahumibe, A. Akinpelu, O. Ayansola, O. Babatunde, A.A.O., C.C., N.G.M., E.C.O., O. Olisa, O.K.A., I.E.N., M.A.E., E.N., R.L.E., R.O.D., A.A., E.O., V.O., C.K.O., S.O., D.I., J.A.A., M.O.A., O.O., O.O., O.K.A., I.E.N.,

M.A.E., E.N., R.L.E., R.O.D., A.A., E.O., V.O., C.K.O., S.O., D.I., E.O.O., N.A.A., C.N.U., K.N.U., N.I.U., C.A., N.A., O. Ayodeji, A.A.L., R.O.I., G.G., A.F., contributed to detection of SARS-CoV-2 samples and selection of positive samples for sequencing. Sequencing analysis and figures were generated by I.B. Olawoye, P.E.O., E.P., D.P. Travel data were provided by K.K. B.A.P., B.L.M., K.J.S., A.Fowotade, S.O., P.O.O., G.A., C.O.E., P.C.S., C.T.H., critically reviewed and edited the manuscript. All authors read and approved the final version of the manuscript.

## Competing interests

The authors declare no competing interests.

## Additional information

Idowu B. Olawoye [1,2,35], Paul E. Oluniyi [1,2,35], Judith U. Oguzie [1,2,35], Jessica N. Uwanibe[1,2,35], Tolulope A. Kayode [1,2,35], Testimony J. Olumade [1,2,35], Fehintola V. Ajogbasile[1,2,35], Edyth Parker [3,35], Philomena E. Eromon[2], Priscilla Abechi[1,2], Tope A. Sobajo[1,2], Chinedu A. Ugwu [1,2], Uwem E. George [1,2], Femi Ayoade [1,2], Kazeem Akano[1,2], Nicholas E. Oyejide[2], Iguosadolo Nosamiefan[2], Iyanuoluwa Fred-Akintunwa[2], Kemi Adedotun-Sulaiman[2], Farida B. Brimmo[2], Babatunde B. Adegboyega[2], Courage Philip[2], Richard A. Adeleke [2], Grace C. Chukwu[2], Muhammad I. Ahmed [2], Oludayo O. Ope-Ewe[2], Shobi G. Otitoola[2], Olusola A. Ogunsanya[2], Mudasiru F. Saibu[2], Ayotunde E. Sijuwola[2], Grace O. Ezekiel[2], Oluwagboadurami G. John[1,2], Julie O. Akin-John[1,2], Oluwasemilogo O. Akinlo[2], Olanrewaju O. Fayemi[2], Testimony O. Ipaye[2], Deborah C. Nwodo[2], Abolade E. Omoniyi[2], Iyobosa B. Omwanghe[2], Christabel A. Terkuma[2], Johnson Okolie[2], Olubukola Ayo-Ale[2], Odia Ikponmwosa[4], Ebo Benevolence[4], Grace O. Naregose[4], Akhilomen E. Patience[4], Osiemi Blessing[4], Airende Micheal[4], Agbukor Jacqueline[4], John O. Aiyepada[4], Paulson Ebhodaghe[4], Omiunu Racheal[4], Esumeh Rita[4], Giwa E. Rosemary[4], Ehikhametalor Solomon[4], Ekanem Anieno[4], Yerumoh Edna[4], Aire O. Chris[4], Adomeh I. Donatus[4], Ephraim Ogbaini-Emovon[4], Mirabeau Y. Tatfeng[5], Hannah E. Omunakwe [6], Mienye Bob-Manuel[6], Rahaman A. Ahmed [7], Chika K. Onwuamah [7], Joseph O. Shaibu[7], Azuka Okwuraiwe [7], Anthony E. Ataga[8], Andrew Bock-Oruma[9], Funmi Daramola[10], Ibrahim F. Yusuf[10], Akinwumi Fajola[11], Nsikak-Abasi Ntia[12], Julie J. Ekpo[13], Anietie E. Moses [13], Beatrice W. Moore-Igwe[14], Oluwatosin E. Fakayode[15], Monilade Akinola[16], Ibrahim M. Kida[17], Bamidele S. Oderinde [17], Zara W. Wudiri[18], Oluwapelumi O. Adeyemi [19], Olusola A. Akanbi [20], Anthony Ahumibe[20], Afolabi Akinpelu [20], Oyeronke Ayansola[20], Olajumoke Babatunde[20], Adesuyi A. Omoare [20], Chimaobi Chukwu [20], Nwando G. Mba[20], Ewean C. Omoruyi[21], Olasunkanmi Olisa[22], Olatunji K. Akande [22], Ifeanyi E. Nwafor[23], Matthew A. Ekeh[23], Erim Ndoma[23], Richard L. Ewah[23], Rosemary O. Duruihuoma[23], Augustine Abu[23], Elizabeth Odeh[23], Venatius Onyia[23], Chiedozie K. Ojide[23], Sylvanus Okoro[24], Daniel Igwe[24], Emeka O. Ogah[24], Kamran Khan[25,26], Nnennaya A. Ajayi [27], Collins N. Ugwu[27], Kingsley N. Ukwaja[27], Ngozi I. Ugwu[28], Chukwuyem Abejegah[29], Nelson Adedosu[29], Olufemi Ayodeji[29], Ahmed A. Liasu[29], Rafiu O. Isamotu[30], Galadima Gadzama[31], Brittany A. Petros [32], Katherine J. Siddle [32], Stephen F. Schaffner [32], George Akpede[4], Cyril Oshomah Erameh[4], Marycelin M. Baba [16,17], Femi Oladiji [33], Rosemary Audu[7], Nnaemeka Ndodo[20], Adeola Fowotade[23], Sylvanus Okogbenin[4], Peter O. Okokhere[4], Danny J. Park [32], Bronwyn L. Mcannis [32], Ifedayo M. Adetifa[20], Chikwe Ihekweazu[20], Babatunde L. Salako[7], Oyewale Tomori [2], Anise N. Happi[2], Onikepe A. Folarin [1,2], Kristian G. Andersen [3], Pardis C. Sabeti [32,34] & Christian T. Happi [1,2,34] ✉

[1]Department of Biological Sciences, Faculty of Natural Sciences, Redeemer's University, Ede, Osun State, Nigeria. [2]African Centre of Excellence for Genomics of Infectious Diseases (ACEGID), Redeemer's University, Ede, Osun State, Nigeria. [3]Department of Immunology and Microbiology, The Scripps Research Institute, La Jolla, CA, USA. [4]Irrua Specialist Teaching Hospital, Irrua, Edo State, Nigeria. [5]Department of Medical Laboratory Science, Niger Delta University, Amassoma, Bayelsa State, Nigeria. [6]Satellite Molecular Laboratory, Rivers State University Teaching Hospital, Port Harcourt, Rivers State, Nigeria. [7]The Nigerian Institute of Medical Research, Yaba, Lagos State, Nigeria. [8]Molecular Laboratory, Regional Centre for Biotechnology and Bioresources Research,

University of Port Harcourt, Port Harcourt, Rivers State, Nigeria. [9]Family Physician, SPDC, Port Harcourt, Rivers State, Nigeria. [10]Clinical Health, SPDC, Port Harcourt, Rivers State, Nigeria. [11]Regional Community Health, SPDC, Port Harcourt, Rivers State, Nigeria. [12]Occupational Health, SPDC, Port Harcourt, Rivers State, Nigeria. [13]Department of Medical Microbiology and Parasitology, University of Uyo, Uyo, Akwa Ibom State, Nigeria. [14]Rivers State University, Port Harcourt, Rivers State, Nigeria. [15]Department of Public Health, Ministry of Health, Ilorin, Kwara State, Nigeria. [16]WHO Polio Laboratory, University of Maiduguri Teaching Hospital, Maiduguri, Borno State, Nigeria. [17]Department of Immunology, University of Maiduguri Teaching Hospital, Maiduguri, Nigeria. [18]Department of Community Medicine, University of Maiduguri Teaching Hospital, Maiduguri, Borno State, Nigeria. [19]Department of Medical Microbiology and Parasitology. Faculty of Basic Clinical Sciences. College of Health Sciences, University of Ilorin, Ilorin, Kwara State, Nigeria. [20]Nigeria Centre for Disease Control, Abuja, Nigeria. [21]Medical Microbiology and Parasitology Department, College of Medicine, University of Ibadan, Ibadan, Nigeria. [22]Biorepository Clinical Virology Laboratory, University of Ibadan, Ibadan, Nigeria. [23]Virology Laboratory, Alex Ekwueme Federal University Teaching Hospital, Abakaliki, Ebonyi State, Nigeria. [24]Alex Ekwueme Federal University Teaching Hospital, Abakaliki, Nigeria. [25]Department of Medicine, University of Toronto, Toronto, Canada. [26]BlueDot, Toronto, Canada. [27]Internal Medicine Department, Alex Ekwueme Federal University Teaching Hospital, Abakaliki, Nigeria. [28]Haematology Department, Alex Ekwueme Federal University Teaching Hospital, Abakaliki, Nigeria. [29]Federal Medical Center, Owo, Ondo State, Nigeria. [30]Ministry of Health, Osogbo, Osun State, Nigeria. [31]Department of Medical Microbiology, University of Maiduguri Teaching Hospital, Maiduguri, Borno State, Nigeria. [32]Broad Institute of Harvard and MIT, Cambridge, MA, USA. [33]Department of Epidemiology and Community Health, Faculty of Clinical Sciences, College of Health Sciences, University of Ilorin, Ilorin, Nigeria. [34]Department of Immunology and Infectious Diseases, Harvard TH Chan School of Public Health, Boston, MA, USA. [35]These authors contributed equally: Idowu B. Olawoye, Paul E. Oluniyi, Judith U. Oguzie, Jessica N. Uwanibe, Tolulope A. Kayode, Testimony J. Olumade, Fehintola V. Ajogbasile, Edyth Parker. ✉e-mail: happic@run.edu.ng

