## [Peer Review File · Nature Communications]

Emergence and spread of two SARS-CoV-2 variants of interest in NigeriaEditorial Note: This manuscript has been previously reviewed at another journal that is not operating a transparent peer review scheme. This document only contains reviewer comments and rebuttal letters for versions considered at *Nature Communications*. Mentions of prior referee reports have been redacted.

REVIEWER COMMENTS

Reviewer #1 (Remarks to the Author):

The authors have substantially revised the manuscript. The analysis for B.1.1.318 provides fresh conclusions about the lineage's origins, which raises some additional questions. Is the location "Africa" overrepresented in the B.1.1.318 tree? If so, could this affect the inference of the root state and merit "Africa" being downsampled or broken into smaller sub-categories (southern Africa, northern Africa, east Africa, west Africa, etc.)? Or "South Africa" and "other countries in Africa"? How you cut the data has a big effect on the root inference.

[REDACTED]

Reviewer comment: [REDACTED]. The state of Borno in the northeast does not appear to be undersampled (the radius of the circle is large). Yet there is a very different strain composition in this state. There is also a different strain composition in FCT Abuja. It seems worth commenting on some of the within-country dynamics in Nigeria and how the southern states seem to be linked and have similar strain composition (presumably reflecting high viral gene flow) and states farther away geographically are less linked and show different strain dynamics.

Minor comments:

I agree with other reviewers that the colors in the tree are difficult to read. It makes sense to include a few locations of interest and group the other locations that are not mentioned in the text (e.g., SE Asia, South America, Southern Asia, Middle East, Central America, Central Asia) as "other"?

Line 251: "North America"

Line 111: do you mean "undersampled"?

Lines 111-114: These final conclusions in the abstract are vague (could be written for any genomic epidemiology paper from any country). Could you write something more specific to what this study found in Nigeria/Africa?

Reviewer #2 (Remarks to the Author):

This is an interesting paper on the phylogenetic origins of the B.1.525 and B.1.1.318 lineages of SARS-CoV-2, both of which passed through Nigeria's impressive genomic surveillance system. The paper presents substantial new country-specific genomic data (assuming these were not already

submitted to GenBank/GISAID) and some limited epidemiological analysis based on the introduction "III" index and flight data. The best supported results in the paper are on the phylogeographic origins of the two lineages that are studied, but I don't think that these results are foundational enough or important enough to SARS-CoV-2 evolution to warrant publication in Nat Commun. One obvious reason is that these two lineages have stopped circulating. A second reason is that there are already many publications and regular updates on NextStrain, Pango, and other platforms on this topic.

The origins analyses in Figures 2A and 3A seem to be well done, but these analyses are now routine. Was it not known already that the Eta variant came from Europe? None of the variant analyses/reports shown in NextStrain or Pango had descriptions of Eta evolution and where it came from? (Also, if you're focusing on a variant origins analysis, it is worthwhile to check that there is no recombination in your data set .. in the early data sets, especially country-specific data sets, there was little recombination, so you probably have little to worry about). This 'variant origins' part of the paper is publishable and better suited to Virus Evolution or Mol Biol Evol, but perhaps not a general science journal like Nat Commun.

The links to travel restrictions are mentioned in many parts of the results, but you don't know that the virus emerged in one place because of travel restrictions/relaxations; you don't know that the virus was/wasn't imported because of travel restrictions/relaxations. These could just be coincidences, as there are only several periods of travel to look at.

The III and EII are based on time series of deaths. But this assumes that (1) death numbers in Nigeria are complete (are they?) and (2) that the ratio of deaths-to-asymptomatic cases is the same in Nigeria as in Europe. Assumption (2) can't possibly be true because the age structure in Africa is much younger resulting in fewer deaths and more asymptomatic cases.

The fact that there is strong regional connectivity (in terms of migration) and strong international connectivity with Europe when flights are not restricted is not surprising. But I don't think this is a publishable result.

Reviewer #3 (Remarks to the Author):

I believe that the authors have adequately addressed comments from the previous round of reviews. The only concern that I still have is that the authors say that one of the sources of sequence data for this analysis were samples from "in-bound or out-bound travellers who had to take mandatory COVID-19 tests". Given that one of the main focuses of this analysis was phylogeographic analysis, these sequences could bias its interpretations. For example, if testing was required from one location (i.e. UK because of the emergence of Alpha), but not the other, then you are more likely to identify introductions from the UK than other locations. Please investigate if the findings hold without those sequences. Alternatively, if it is impossible to identify whether sequences come from travellers or from other sources, this bias should be discussed and authors should indicate which countries were on/off the list of countries that required travellers to be tested prior to arriving to Nigeria.

Minor comments:

Line 111 – did you mean “undersampled” or “underrepresented”

Line 227 – Please make sure you specify year, not only months throughout the manuscript

Reviewer #1

The authors have substantially revised the manuscript. The analysis for B.1.1.318 provides fresh conclusions about the lineage's origins, which raises some additional questions. Is the location "Africa" overrepresented in the B.1.1.318 tree? If so, could this affect the inference of the root state and merit "Africa" being downsampled or broken into smaller sub-categories (southern Africa, northern Africa, east Africa, west Africa, etc.)? Or "South Africa" and "other countries in Africa"? How you cut the data has a big effect on the root inference.

We thank the reviewer for their consideration and helpful comments. We agree with the reviewer that phylogeographic work is fundamentally subject to sampling biases. As other's work has shown^{1,2}, models that explicitly incorporate travel history and unsampled lineages from undersampled populations are required to account for the rapid rate of viral dispersal and uneven sampling between locations that may result in unsampled intermediary locations. As we did not have access to travel information, we attempted to ensure evenness and representativeness in our dataset. On a country-level, only ten countries had more than 30 B.1.1.318 sequences available at the time of the study. We randomly downsampled all ten countries with more than 30 sequences to 30, which was 10 sequences above the mean (Figure 2). As per Figure 2, the region state Europe included twice as many sequences as Africa after downsampling on a country-level, indicating that the tree's most probable root state Africa is not likely driven by oversampling.

Figure 2: Sampling numbers by region for B.1.1.318 phylogeography

[REDACTED]

Reviewer comment: [REDACTED] The state of Borno in the northeast does not appear to be undersampled (the radius of the circle is large). Yet there is a very different strain composition in this state. There is also a different strain composition in FCT Abuja. It seems worth commenting on some of the within-country dynamics in Nigeria and how the southern states seem to be linked and have similar strain composition (presumably reflecting high viral gene flow) and states farther away geographically are less linked and show different strain dynamics.

We are grateful for reviewer 1's comment. Majority of the Northern states were not adequately sampled during the third wave except for Kano and Plateau states. Borno state, FCT Abuja and others were sampled during the first two waves. Hence the reason we stated the sampling limitation in the study "The sampling gap was the inability to sequence samples from every state during all different waves of the pandemic in Nigeria" in line 415. Also we added a text in line 167 to highlight that in our results "The northern states were highly undersampled relative to the southern states, with sampling highly uneven across time especially during the third wave". The radii in the south are considerably larger and more in proportion compared to the north. We have also included that in our discussion as one of the study's limitations (line 395).

Minor comments:

I agree with other reviewers that the colors in the tree are difficult to read. It makes sense to include a few locations of interest and group the other locations that are not mentioned in the text (e.g., SE Asia, South America, Southern Asia, Middle East, Central America, Central Asia) as "other"?

We agree and have adapted the Figures 2 and 3 as well as indicating in the figure legend that "Locations with negligible contributions have been grouped and annotated in gray."

Line 251: "North America"

We have corrected the text from Northern America to North America in line 253.

Line 111: do you mean "undersampled"?

Yes, and we have now corrected this to undersampled in line 111.

Lines 111-114: These final conclusions in the abstract are vague (could be written for any genomic epidemiology paper from any country). Could you write something more specific to what this study found in Nigeria/Africa?

We agree with reviewer 1 and we have included a more descriptive text about our findings that is peculiar to our study which reads "In this study, we see how regional connectivity of Nigeria drove the spread of these variants of interest to surrounding countries and those connected by air-traffic" in line 112.

Reviewer #2

This is an interesting paper on the phylogenetic origins of the B.1.525 and B.1.1.318 lineages of SARS-CoV-2, both of which passed through Nigeria's impressive genomic surveillance system. The paper presents substantial new country-specific genomic data (assuming these were not already submitted to GenBank/GISAID) and some limited epidemiological analysis based on the introduction "III" index and flight data. The best supported results in the paper are on the phylogeographic origins of the two lineages that are studied, but I don't think that these results are foundational enough or important enough to SARS-CoV-2 evolution to warrant publication in Nat Commun. One obvious reason is that these two lineages have stopped circulating. A second reason is that there are already many publications and regular updates on NextStrain, Pango, and other platforms on this topic.

The origins analyses in Figures 2A and 3A seem to be well done, but these analyses are now routine. Was it not known already that the Eta variant came from Europe? None of the variant analyses/reports shown in NextStrain or Pango had descriptions of Eta evolution and where it came from? (Also, if you're focusing on a variant origins analysis, it is worthwhile to check that there is no recombination in your data set .. in the early data sets, especially country-specific data sets, there was little recombination, so you probably have little to worry about). This 'variant origins' part of the paper is publishable and better suited to Virus Evolution or Mol Biol Evol, but perhaps not a general science journal like Nat Commun.

The links to travel restrictions are mentioned in many parts of the results, but you don't know that the virus emerged in one place because of travel restrictions/relaxations; you don't know that the virus was/wasn't imported because of travel restrictions/relaxations. These could just be coincidences, as there are only several periods of travel to look at.

We note the reviewer's concerns. However, we clarify that we're quantifying introduction *risk* when discussing the connectivity data and import/export indices, not sampled introductions. We have been careful to frame the results in terms of *risks* e.g. the lines "We sought to investigate this further by quantifying the introduction *risk*" and "This suggests that the introduction *risk* was predominantly driven by regional connectivity during the period when travel restrictions were in place" and "we found that the introduction *risk* from air travel was very low during the period of travel restrictions" and "In our III analyses, we found negligible introduction *risk* from outside of Europe and North America." We do not think that quantification and discussion of *risk* translates to discussing causality and have read the text to ensure we frame the discussion as a matter of estimated *risk* rather than sampled viral introductions throughout.

The III and EII are based on time series of deaths. But this assumes that (1) death numbers in Nigeria are complete (are they?) and (2) that the ratio of deaths-to-asymptomatic cases is the same in Nigeria as in Europe. Assumption (2) can't possibly be true because the age structure in Africa is much younger resulting in fewer deaths and more asymptomatic cases.

We agree this needs clarification and thank the reviewer. We note that these estimates quantify the temporal trend in the daily estimated number of introductions or exports rather than absolute values, as it is based on the back-extrapolated time series of deaths and is therefore restricted by the associated reporting delays and biases. We have clarified how we interpret the indices in our Discussion in lines 379 to 384. Towards addressing the Reviewer 2's concern on the effect of varying ratios of asymptomatic infections to deaths in Nigeria in the parametrization of the EII, we have conducted sensitivity analyses with the proportion of asymptomatic infections fixed at 0.1 and 0.5 (as opposed to the global average of 0.3 used in the main text). As per Figure 1, this has not changed our inference, as we interpret the EII and III as temporal trends, not measures of absolute risk, as we clarified in the main text (line 385). We have also included the text “, including Nigeria, “ in line 390 to indicate that our statements on the effect of case underascertainment from modeled deaths on export and introduction intensity indexes extends to Nigeria as well. The restructuring of this paragraph now includes both these statements, as well as the text previously included in the manuscript which states the likely effect missing land-based travel will have on underestimation of risk estimates relating to neighbouring countries:

“The III and EII indices should be interpreted to quantify the temporal trend in the daily estimated number of introductions or exports rather than absolute values. The indices are based on the back-extrapolated time series of deaths and are therefore restricted by the associated reporting delays and biases. Notably, they will be underestimated if there is large-scale underascertainment in source countries, as previously shown for African, including Nigeria, based on excess mortality data. Most, notably our import risk index underestimates the importance of regional connectivity in driving introduction estimates, as they likely result from shorter distance connectivity to surrounding countries, which we could not collect data on”

Figure 1: *Estimated Exportation Intensity Index (EII) for Nigeria from 2020-2021 with varying proportion of asymptomatic infection A) 0.1 and B) 0.5.*

The fact that there is strong regional connectivity (in terms of migration) and strong international connectivity with Europe when flights are not restricted is not surprising. But I don't think this is a publishable result.

Reviewer #3

I believe that the authors have adequately addressed comments from the previous round of reviews. The only concern that I still have is that the authors say that one of the sources of sequence data for this analysis were samples from “in-bound or out-bound travelers who had to take mandatory COVID-19 tests”. Given that one of the main focuses of this analysis was phylogeographic analysis, these sequences could bias its interpretations. For example, if testing was required from one location (i.e. UK because of the emergence of Alpha), but not the other, then you are more likely to identify introductions from the UK than other locations. Please investigate if the findings hold without those sequences. Alternatively, if it is impossible to identify whether sequences come from travelers or from other sources, this bias should be discussed and authors should indicate which countries were on/off the list of countries that required travelers to be tested prior to arriving in Nigeria.

We thank reviewer 3 for their insightful comments. Every passenger visiting or returning to Nigeria was mandated to undertake the COVID-19 PCR test upon arrival to Nigeria with no country exception. In addition, there isn't a way to differentiate between in-bound, out-bound and community tested samples as they were not stated in the sample manifest that accompanied the samples. We have included this limitation in the discussion section in line 364.

Minor comments:

Line 111 – did you mean “undersampled” or “underrepresented”

We meant undersampled and have corrected this to reflect that in line 111

Line 227 – Please make sure you specify year, not only months throughout the manuscript

Thanks for pointing that out, we have specified the year throughout the manuscript as seen in lines 228, 231, and 287.

1. Butera, Y. *et al.* Genomic sequencing of SARS-CoV-2 in Rwanda reveals the importance of incoming travelers on lineage diversity. *Nat. Commun.* **12**, 5705 (2021).
2. Lemey, P. *et al.* Accommodating individual travel history and unsampled diversity in Bayesian phylogeographic inference of SARS-CoV-2. *Nat. Commun.* **11**, 5110 (2020).